# Determinants of Suicide Ideation in the Romanian Population

**DOI:** 10.3390/ijerph191610384

**Published:** 2022-08-20

**Authors:** Mihaela Elvira Vuscan, Cristina Faludi, Sergiu Ionica Rusu, Mihaela Laura Vica, Stefana Balici, Costel Vasile Siserman, Horia George Coman, Horea Vladi Matei

**Affiliations:** 1Department of Cellular and Molecular Biology, Iuliu Haṭieganu University of Medicine and Pharmacy, 400349 Cluj-Napoca, Romania; 2Legal Medicine Institute, 400006 Cluj-Napoca, Romania; 3Department of Social Work, “Babeṣ Bolyai” University, 400604 Cluj-Napoca, Romania; 4Doctoral School of Sociology, “Babeṣ Bolyai” University, 400604 Cluj-Napoca, Romania; 5Department of Legal Medicine, Iuliu Haṭieganu University of Medicine and Pharmacy, 400349 Cluj-Napoca, Romania; 6Department of Medical Psychology, Iuliu Haṭieganu University of Medicine and Pharmacy, 400349 Cluj-Napoca, Romania

**Keywords:** suicide ideation, psychosocial determinants, prolonged sadness, quality of life, loneliness

## Abstract

Background: Suicide ideation and behaviors are directly linked to the risk of death by suicide. In Romania, as well as worldwide, increased suicide rates were observed in the recent past, more so in the context of the COVID-19 pandemic. The purpose of this study was to investigate the influence of psychosocial factors, quality of life (QOL), and loneliness dimensions and adverse life antecedents on suicide ideation (SI) and prolonged sadness (PS). Methods: This cross-sectional quantitative research study used a CATI data gathering method to investigate 1102 randomly selected individuals over 18 years of age regarding various determinants of SI and PS. Data were collected in June 2021. Descriptive, inferential, and multivariate statistics were used for data analysis. Results: SI was negatively correlated with all the assessed psychosocial factors, more significantly with family relationships, wealth, health, social relationships, and affective life. Stronger correlations were observed when investigating the state of prolonged sadness, sex, and affective lives along with health and income, which were more influential. SI was negatively correlated with QOL and positively correlated with adverse life events and total loneliness scores. Lesser educated youngsters with reduced overall happiness and a history of depression, self-harm, and trauma were at greater risk of developing SI. Conclusions: This is the first national study exploring the suicide ideation and prolonged sadness in relation to psychosocial factors, quality of life, and adverse life events. These results have important implications for suicide prevention programs, which should be designed in accordance with similar studies.

## 1. Introduction

Despite being a preventable cause of death, suicide claims yearly a toll of 700,000 victims worldwide. It affects elders and youngsters of both high- and low-income countries, the latter to a much greater extent [1].

In Romania, suicide rates have varied from 10.9 to 17.1 over the last four decades, except for a minimum of 9.4 in 1990, after the overthrow of the communist regime [2,3].

As suicide attempters are presenting the highest risk of death by suicide [1], identifying persons experiencing suicide ideation and behaviors is the focus of prevention strategies, with numerous research studies investigating risk factors in view of developing clinical interventions and public health politics [4].

Suicidal ideation (SI) describes wishes and preoccupations with death. Various definitions include or ignore planning deliberations, with no gold standard for managing people at risk of suicide being useful in identifying those whose death by suicide is imminent. The magnitude and characteristics of SI fluctuate dramatically from one person to another or in the same person over (sometimes brief) periods of time [5]. A complex phenomenon conditioned by a multitude of medical, social, and environmental factors, the suicidal act is often triggered by acute stressors [6].

Most studies on SI and suicidal behavior (SB) focus on demographic, social, and medical risk factors, the latter including internal (hopelessness, anxiety, mood disorders, etc.) and external psychopathology (impulsivity, aggressive behaviors, substance abuse, etc.) or a history of suicidal antecedents (previous self-harm thoughts or behaviors). Socio-demographic factors more frequently associated with SI are age, gender, marital status, education, socio-economic status, employment, ethnicity, religion). Socio-emotional factors to be considered in this context include loneliness, social isolation, poor connectivity with family or friends, cognitive inhibition, psychological pain, stressful life events, or abuse history [4,5,6,7].

Subjective QOL is a standard health outcome measure dealing with physical and social dimensions, role functioning, mental and general health of an individual, its assessment serving as an indicator of subjective well-being [8]. The QOL is reflected by the individual’s nonspecific perception in relation to their goals, standards, expectations, and concerns in the context of their culture and value system [9].

Since Durkheim associated suicide with social integration and connectedness in 1897, a plethora of theories highlighted the relevance of social variables on this devastating health problem [10].

The relationships between depression or other mental illnesses and SI or SB are well documented [8,11,12,13]. Social isolation, loneliness, lack of connectedness, and social support have been reported as risk factors for developing suicidal thoughts and behaviors not only for elders, but also for adolescents or various categories of patients [10]. While social isolation is an objective measure of scarce social interactions, loneliness is a subjective perception of being isolated. A distinction is made between emotional loneliness (absence of a close relationship) and social loneliness (lack of a broad social network) [14]. A 2019 review found that the main social constructs associated with suicidal outcomes were living alone, social isolation, loneliness, alienation, and belongingness [10].

A series of recommendations adopted by the European Parliament (resolution 2008/2209 on Mental Health) emphasize the need to implement national programs for the prevention of depression and suicide in all member states [15]. The recent COVID-19 pandemic adds an additional challenge to global mental health issues, triggering a significant increase in the prevalence of anxiety and depression worldwide [16].

This study aimed to explore a set of psychosocial factors, global QOL indicators, items of subjective socio-emotional loneliness, and life antecedents as determinants of suicide ideation and prolonged sadness. The main research questions were: (1) which items of each explanatory factor correlate with suicide ideation and prolonged sadness, and (2) which factors have greater impact on suicide ideation and prolonged sadness?

## 2. Materials and Methods

### 2.1. Study Design and Participants

This study included a cross-sectional quantitative research design. Data were collected between 28 and 30 June 2021, using the computer-assisted telephone interviewing [CATI] method [17]. Data gathering was conducted by the Romanian Institute for Evaluation and Strategy. A sample of 1102 individuals aged 18 and over, representative for the non-institutionalized adult population residing in Romania, was contacted through the random digit dialing (RDD) method. Data were weighted by gender, age, region, area of residence, and level of education, against the most recent official data available (the 2019 population estimates provided by the National Institute of Statistics in Romania, plus level of education data from the 2011 census), using the rake weighting procedure with 20 iterations and a delta of 0.5 in SPSS, version 22, IBM, Cluj-Napoca, Romania.

### 2.2. Ethical Issues

The study was approved by the Ethics Commission of the ‘Iuliu Hațieganu’ University of Medicine and Pharmacy in Cluj-Napoca (No. 270/30 July 2019). The presentation script used by the interview operators included several mandatory elements: (i) the name of the operator and of the research institute; (ii) the topic of the research; (iii) information related to confidentiality, anonymity, and data protection policy; (iv) invitation to participate in the study; (v) request of a verbally expressed informed consent.

### 2.3. Variables and Measurements

The survey questionnaire included a set of sociodemographic data. For this study only gender, age, education, and area of residence were included in the analysis.

The respondents’ perception on such psychosocial factors was assessed through the question: ‘Considering the last 12 months how do you rate different aspects of your life?’ Ten issues were investigated: (a) health; (b) nutrition; (c) physical activities; (d) recreational activities; (e) wealth; (f) affective life; (g) sex life; (h) family relationships; (i) social relationships; (j) income. A ten-point Likert scale was used to evaluate the set of psychosocial factors, ranging from 0 (very dissatisfied) to 10 (very satisfied) [18].

The questionnaire comprised two questions measuring the overall quality of life, namely: ‘How satisfied are you with your life?’ and ‘How happy are you in general?’ Again, a ten-point Likert scale was used to measure the two items, varying from 0 (very dissatisfied/unhappy) to 10 (very satisfied/happy) [19].

Different life antecedents were explored in the questionnaire. The message addressed to the respondents was: ‘I will now read you a list of possible events. Please tell me if you were personally faced with any of these things in your life.’ From the list of antecedents, the following were selected for this study: trauma; abuse; chronic illness; deception; depression; and self-harm. All items admitted a dichotomous answer: ‘yes’ or ‘no’.

The Short Scale of Socio-Emotional Loneliness applied by De Jong Gierveld and Tilburg [20] was used to assess the perceived loneliness, covering two dimensions: social loneliness (SL) and emotional loneliness (EL), respectively. The scale has 6 interspersed statements about a set of ordinary events that the respondent experienced shortly before the interview. Each item admitted three possible answers: ‘yes’, ‘more or less’, and ‘no’. The second option was also considered an indicator for the presence of loneliness. The items measuring social loneliness were formulated in positive terms, as follows: ‘There are plenty of people I can lean on in case of trouble’, ‘There are enough people that I can count on completely’ and ‘There are enough people that I feel close to’. The items assessing the emotional dimension of loneliness were expressed in negative terms, as follows: ‘I experience a general sense of emptiness’, ‘I miss having people around’, ‘Often, I feel rejected’. None of the items directly referred to loneliness explicitly.

The dependent variables in the present study were suicide ideation and prolonged sadness. They were measured using the following questions: ‘Please tell me if you have ever thought of suicide or considered taking your own life’, and ‘Please tell me if you have ever experienced long-term sadness’. Both questions admitted a dichotomous answer: ‘yes’ or ‘no’ [5].

### 2.4. Data Analysis

Procedures of descriptive, inferential statistics and multivariate analyses were applied to extract answers to the research questions. The Pearson’s correlation test and the binary logistic regression were applied to explore the statistical relationship between the study variables and suicide ideation/prolonged sadness in the Romanian general population. All levels of significance were two-tailed, and significance values less than 0.05 were considered statistically significant. Data were analyzed with the SPSS program, version 22.

## 3. Results

### 3.1. Descriptive Statistics

Out of the total number of respondents, 51% were women (N = 563) and 54% originated from urban areas (N = 594). Regarding the age distribution, 26% were between 18–35 years of age, 29% were in the 36–50 years group, 23% had 51 to 65 years and the remaining 22% were over 65 years. According to the level of education, 37% had primary or lower education, 48% accomplished secondary education, and 15% got a higher education diploma (data not shown in a table). In the investigated national sample, 442 (40%) of respondents reported they experienced prolonged sadness and 90 (8%) confessed they thought about suicide (data not shown in the table).

### 3.2. Inferential Analysis Results

In the investigated national sample, the results of the inferential analysis showed, as expected, that suicide ideation is negatively correlated with all the assessed psychosocial factors (Table 1). In descending order, family relationships (r = −0.222, *p* = 0.01), wealth (r = −0.147, *p* = 0.01), health (r = −0.144, *p* = 0.01), social relationships (r = −0.139, *p* = 0.01), and affective life (r = −0.123, *p* = 0.01) produced the greatest impact on suicidal ideation. The coefficients were even higher when the prolonged sadness was investigated, the psychosocial factors better correlated with a state of prolonged sadness being sex life (r = −0.286, *p* = 0.01), affective life (r = −0.279, *p* = 0.01), wealth (r = −0.262, *p* = 0.01), health (r = −0.238, *p* = 0.01), and income (r = −0.215, *p* = 0.01), to mention just the most influential.

Suicide ideation was negatively correlated with the QOL global indicators and positively correlated with adverse life antecedents, as illustrated in Table 2. Correlation was higher between both suicide ideation and prolonged happiness and overall happiness (r = −0.236, *p* = 0.01, and r = −0.337, *p* = 0.01, respectively) than between both dependent variables and life satisfaction (r = −0.166, *p* = 0.01, and r = −0.241, *p* = 0.01, respectively). Depression (r = 0.360, *p* = 0.01), self-harm (r = 0.317, *p* = 0.01), deception (r = 0.229, *p* = 0.01), and trauma (r = 0.200, *p* = 0.01) were risk factors for suicide ideation and prolonged sadness. The manifestation of a prolonged state of sadness was positively correlated with all the investigated adverse life antecedents: depression (r = 0.416, *p* = 0.01), deception (r = 0.390, *p* = 0.01), trauma (r = 0.256, *p* = 0.01), self-harm (r = 0.243, *p* = 0.01), chronic illness (r = 0.233, *p* = 0.01), and abuse (r = 0.229, *p* = 0.01).

The correlation between the studied dependent variables and the components of the Socio-Emotional Loneliness Scale revealed that two items of the scale play an important role (Table 3). Thus, experiencing a general sense of emptiness (r = 0.271, *p* = 0.01) and often feeling rejected (r = 0.254, *p* = 0.01) are positively correlated with suicide ideation. In a similar manner, the same items were positively correlated with experiencing prolonged sadness (r = 0.308, respectively r = 0.246, *p* = 0.01).

### 3.3. Logistic Regression Results

The logistic regression model is illustrated in Table 4. The dependent variable was the existence of suicidal ideation, admitting a dichotomous answer: 1—yes and 0—no. The main socio-demographic characteristics of the respondents were treated as control variables in the model. Covariates were operationalized as dichotomous variables, including the psychosocial factors, the global indicators for quality of life, the existence of socio-emotional loneliness, and three adverse life events (depression, self-harm and trauma). The results of the multivariate analysis revealed that the younger age groups and the individuals with the lowest levels of education registered the highest risk of confronting with suicidal ideation, the results being highly significant. In contrast, sex and area of residence bore no significant influence on the manifestation of suicide ideation.

Concerning the covariates, individuals with a low level of overall happiness were more exposed to suicide ideation (exp (B) = 3.663, *p* = 0.001). In opposition, persons who had not experienced life events such as depression (exp (B) = 0.130, *p* < 0.001), self-harm (exp (B) = 0.223, *p* < 0.001), and trauma (exp (B) = 0.555, *p* = 0.049) registered a significantly lower risk of being confronted with suicide ideation.

Life satisfaction and the level of perceived socio-emotional loneliness had no significant influence on the manifestation of suicidal ideation.

## 4. Discussion

All determinants investigated in this study (psychosocial factors, global QOL indicators, items of subjective socio-emotional loneliness, life antecedents) were found to be correlated with suicide ideation (SI) and profound sadness (PS). The multivariate analysis focused on the investigation of different possible explanatory factors of SI. The results of the logistic regression showed that younger people are more exposed to suicide ideation. In terms of education, people with a lower level of education presented greater risks of developing suicidal thoughts. Perceiving a low level of overall happiness increased the risk of SI. The regression model also revealed that people who did not experience previous episodes of depression and did not report self-harm and trauma were not confronted with suicide ideation. It is estimated that up to a third of Western country individuals experience suicide ideation at some point during their lives [21]. In the studied national sample, 8% of respondents had experienced suicidal thoughts. The 18–35 years age group registered a 6.8 times higher risk of experiencing SI. This is similar with that of a five-year survey (2009–2014) by the Center for Behavioral Health Statistics and Quality, in which the question ‘At any time in the past 12 months, did you seriously think about trying to kill yourself?’ produced 6% affirmative responses in the 18–25-year-olds and 1.6% in those aged 65 years and above [5]. However, when it comes to suicide, rates are higher among the category of elders. In Romania, the most recent data indicated that in 2019 the crude suicide rate was 14.52 in the age group 65–74 versus 6.3 in the age group 25–34 [22]. Suicide is the second leading cause of death among youngsters and children and suicidal ideation and behavior is even more common in the 10–24-years-old group [23]. In older adults the complex causes of suicide range from economic difficulties, physical or psychological pain, to loneliness and social exclusion, but also include mental disorders, cognitive impairment, or inhibition [7]. As for the youngsters, extensive social media exposure in the internet era is held accountable for the growing social isolation and loneliness, for youngsters in particular [24]. Tracking loneliness changes from childhood to adolescence, researchers found that they indirectly affect self-harm behaviors and suicidal thoughts through depression and externalizing behavior problems [25]. On the other hand, it is demonstrated that social support has positive effects on health and well-being, acting as a protective factor against developing suicidal ideation [26]. One’s psychological health and mental wellbeing is vastly dependent on one’s social relationships [27]. A strong family network is generally the best antidote against loneliness [2,28].

This study explored the correlation of SI and PS with psychosocial determinants in the general population of Romania. The influence of certain psychosocial factors on the emergence of SI and PS has been well documented in the literature [18]. The impact of socio-demographic factors on developing suicidal behaviors was found to be negatively correlated with their prevalence in the studied population [4]. Along with sex and affective life, wealth and health, income was one of the most influential psychosocial determinants of PS and SI in our study. Another study also found that economic issues such as unemployment, along with a diagnosis of clinical depression, early onset of the mental illness and history of self-harm, were major suicide risk factors identified in a group with intellectual disabilities, similar to their findings for the general population [29].

The influence of low life satisfaction or happiness and poor mental health on SI is well documented [21]. On the background of reduced life satisfaction, depression is seen as the most direct factor affecting suicidal ideation [7].

In this study, highly significant negative correlations for SI and profound sadness were found both against QOL and health or emotional determinants (depression, chronic illnesses, trauma, deception, abuse, self-aggression). As expected, depression provided the most powerful associations with SI and QOL determinants. Relatively strong associations with SI were observed for abuse victims, in persons engaged in self-harm or exposed to negative emotions (e.g., deception).

A history of self-injurious thoughts and behaviors was identified as an important risk for suicide in adolescents and young adults [30]. Compared to healthy controls, suicide attempters presented more stressful life events, lower social support, less healthy coping, and poor QOL [31].

Loneliness and social isolation are socio-emotional factors frequently associated with suicide ideation and prolonged sadness. Loneliness, i.e., the subjective perception of being socially or emotionally isolated, was found to be a correlate of depressive symptoms, independent of gender, other demographic factors, multiple psychosocial variables, and social desirability [27]. It is a strong predictor of suicidal ideation along with higher age, lower socio-economic status, generalized anxiety, panic and depression [32]. Similarly, social isolation was strongly associated with earlier mortality rates, higher rates of depression, increased risk for suicide and worse mental health outcomes or overall QOL [33]. In this study, higher scores of total socio-emotional loneliness scale were associated with SI and PS. Similarly, a survey over all US federal states in April 2020 found that 43% of the 1013 respondents felt very lonely during the lockdown, most of them meeting clinical criteria for moderate to severe depression and a third endorsing certain levels of suicide ideation [34]. Highlighting new facets of the factors influencing the development of suicidal behaviors in the Romanian population helps fighting the consequences of this important health problem, easing the transition towards the WHO desiderates of active aging in these life-threatening pandemic times. The current study has several strengths. As far as we know, this is the first national study exploring the suicide ideation and prolonged sadness in relation to psychosocial factors, quality of life, and life antecedents. The findings can be useful to design adequate national strategy aiming to reduce the death by suicide. They also could serve as a starting point for future research focusing on other determinants of suicide ideation and behaviours (e.g., income, alcohol and drug use, self-esteem).

This study has some limitations. The questions addressing the investigation topic related to different time frames. The questions related to the psychosocial factors concerning the last 12 months, overlapping with the period of the COVID-19 pandemic and the restrictions imposed during this period. The questions about adverse life events, suicide ideation and long-term sadness were lifetime questions, while the socio-emotional loneliness scale referred to the recent past. This shortcoming prevented to establish the chronological order of the investigated issues. For instance, it could not be stated whether the answers at the self-assessment of different aspects of life (family relationship, wealth, social relationships, etc.) reflected a prior or subsequent answer at the subjective assessment of the suicide ideation and prolonged sadness. The results of correlations showed that there existed significant associations between most of the studied variables, without pretending any causal relationships between them. In addition, the logistic regression model included a set of independent variables (gender, age group, area of residence, and education), and covariates related to global quality of life and adverse life events which can explain the occurrence of the suicide ideation. Another limit of the study might be that the telephone interviewers had no psychological training. However, no professional assessments were conducted in regard to the suicidal intentions or prolonged sadness and the questions did not refer to the present situation. Thus, statistical analyses were based on the respondents’ self-assessment alone.

## 5. Conclusions

States of perceived loneliness and rejection induced by social isolation, corroborated with prolonged sadness, lead to depression—a major risk factor for developing suicidal behaviors. This first national study draws attention to the importance that needs to be given to precarious aspects of life easily noticeable to health professionals, family members or entourage which could divert one off a suicidal path. Psychosocial factors, global QOL indicators, adverse life events and indices of subjective loneliness were analyzed in relation to their association with suicidal ideation and/or prolonged sadness. The profile of suicide ideators reflects the following characteristics: young and middle-aged adults with a low level of education, reduced overall happiness, and with a history of depression, trauma, and self-harm. Suicide ideation was found to be independent of gender and area of residence, although great disparities among men and women and between urban and rural areas were well documented in the Romanian population. These results have important implications for suicide prevention programs, which should be designed in accordance with similar research.

Three decades of social change did not override the Romanians’ hostility to address costly professionals for insanity-labeled mental health problems induced by suicidal ideation and prolonged sadness, dismissed as non-existent under the communist regime. In the context of increased loneliness and QOL deterioration under the pandemic and the imminent economic crisis, the greatest responsibility for identifying such problems and short-term interventions rests with family members assisted by health professionals. Family members should take appropriate measures to prevent withdrawal or self-isolation that may lead to suicidal ideation, such as immediate referral to psychotherapy. Using the easy-to-apply data collection tools proposed by this study and considering its results, health professionals can identify people at risk of developing suicidal behaviors; elders and youngsters in particular.

Education initiatives in schools and media exposure of their costly socio-economic consequences should help timely identification and prevention of cross-generational long-term sadness and suicidal ideation. Guidelines providing tools to address unhappy, dissatisfied, depressive, and self-aggressive individuals at risk, and for early intervention protocols, are needed to fight this serious health problem in view of implementing the WHO’s active aging policy allowing a constantly aging population to fulfill its potential for physical, social, and mental wellbeing.

## Figures and Tables

**Table 1 ijerph-19-10384-t001:** Correlations between suicide ideation/prolonged sadness and psychosocial factors.

Variable	Psychosocial Determinants of Suicide												
	Prolonged Sadness	Suicide Ideation	Health	Nutrition	Physical Activities	Recreational Activities	Wealth	Affective life	Sex Life	Family Relationships	Social Relationships	Income	Index of Psychosocial Factors
Suicide ideation	0.289 **	1	−0.144 **	−0.123 **	−0.025	−0.026	−0.147 **	−0.123 **	−0.077 *	−0.222 **	−0.139 **	−0.102 **	−0.165 **
Prolonged sadness	1	0.289 **	−0.238 **	−0.169 **	−0.093 **	−0.092 **	−0.262 **	−0.279 **	−0.286 **	−0.185 **	−0.135 **	−0.215 **	−0.306 **

** Correlation is significant at the 0.01 level (2-tailed). * Correlation is significant at the 0.05 level (2-tailed).

**Table 2 ijerph-19-10384-t002:** Correlations between suicide ideation/prolonged sadness, and quality of life dimension/life antecedents.

Variable	Quality of Life Items		Life Antecedents					
	Life Satisfaction	Overall Happiness	Trauma	Abuse	Chronic Illnesses	Deception	Depression	Self-harm
Suicide ideation	−0.166 **	−0.236 **	0.200 **	0.166 **	0.121 **	0.229 **	0.360 **	0.317 **
Prolongued sadness	−0.241 **	−0.337 **	0.256 **	0.229 **	0.233 **	0.390 **	0.416 **	0.243 **

** Correlation is significant at the 0.01 level (2-tailed).

**Table 3 ijerph-19-10384-t003:** Correlations between suicide ideation/prolonged sadness, and socio-emotional loneliness scale items.

Variable	1	2	3	4	5	6	7
Suicide ideation	0.271 **	−0.017	0.254 **	−0.007	0.097 **	0.080 **	0.203 **
Prolongued sadness	0.308 **	−0.028	0.246 **	0.039	0.138 **	0.105 **	0.255 **

** Correlation is significant at the 0.01 level (2-tailed).Legend. 1: I experience a general sense of emptiness. 2: I miss having people around. 3: I often feel rejected. 4: There are plenty of people I can lean on in case of trouble. 5: There are many people that I can count on completely. 6: There are enough people that I feel close to. 7: Total socio-emotional loneliness score.

**Table 4 ijerph-19-10384-t004:** The results of the logistic regression on the existence of suicidal ideation.

Control Variables	Variables’ Categories	Frequency	%	Exp(B)	Sig.	95% C.I.for EXP(B)	
						Lower	Upper
Sex	Male	546	51	0.648	0.123	0.373	1.125
	Female (ref.)	522	49				
Age-group	18–35 years	277	26	**6.828**	**<0.001**	2.675	17.429
	36–50 years	297	28	**5.372**	**<0.001**	2.195	13.149
	51–65 years	327	30	**4.697**	**<0.001**	1.954	11.291
	Over 65 years (ref.)	167	16				
Education	Primary or lower education	163	15	**3.104**	**0.015**	1.243	7.749
	Secondary education	553	52	1.128	0.798	0.448	2.837
	Higher education (ref.)	352	33				
Area of residence	Urban	611	57	0.787	0.386	0.458	1.353
	Rural (ref.)	457	43				
**Covariates**							
Life satisfaction	Low	624	58	0.904	0.772	0.456	1.790
	High (ref.)	444	42				
Overall happiness	Low	614	57	**3.004**	**0.004**	1.430	6.311
	High (ref.)	454	43				
Depression	No	841	79	**0.152**	**<0.001**	0.085	0.272
	Yes (ref.)	227	21				
Self-harm	No	1009	94	**0.259**	**0.001**	0.135	0.495
	Yes (ref.)	59	6				
Trauma	No	718	67	**0.562**	**0.050**	0.314	1.006
	Yes (ref.)	350	33				

## Data Availability

Not applicable.

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
