# Peer review of "Determinants of Suicide Ideation in the Romanian Population"

_ijerph, 2022, doi:10.3390/ijerph191610384_

Round 1

Reviewer 1 Report

The present manuscript is a national survey conducted in Romania, in which the authors attempt to identify variables predicting which patients are at high risk for suicidal ideation. The manuscript does not present severe English language mistakes, but the overall readability and style must be improved. The sentences in the introduction and discussion are poorly connected. Connecting conjunctions between sentences are missing, making the text lack smoothness. I suggest the authors ask for support from an English native speaker. Moreover, the introduction and the discussion might be shortened. Here are my requests:

Methods

- Please provide a reference to the CATI method

- Please better describe the procedure for weighting, in statistical analysis

- Please better describe if you applied or not multiple comparison corrections, and if not, report it as a limit and try to justify this choice

- The reference you cite in line 178 (for The Short Scale of Socio-Emotional Loneliness) has been translated into Francoise, German, Dutch, Russian, Bulgarian, Georgian, and Japanese. Is there any non-reported validation in the Romanian language? Did you adopt the standard procedure for forward and backward translation if not? It should be clearly stated as a study limitation if you did not. Moreover, it would be precious if at least the Chronbach alpha of the scale in the present study is reported.

- Regarding the question of Suicidal Ideation and long-term sadness, I have two questions:

1. What is the time frame you considered for the questions? Because from the text, those seem lifetime questions. If those are lifetime questions, how could you use the variables as the outcome variables of present suicidal ideation?

2. On the other hand, if you defined a strict and recent timeframe, the question is linked to suicidal risk if the participant answered yes. So then, the second question is more ethical. How did you manage participants' suicidal ideation? Did you suggest participants contact a specific number (suicide prevention lifeline) or go to the hospital seeking help?

Results

- The correlation in table 1 reports a 0.05 level of significance. The others 0.01. Is the difference based on something? Did you apply multiple comparison corrections for the following tables? In case, why did you not in the first one?

- There is some text in bold in the tables, but the reason is not clarified in the caption of the table. Please, clarify.

- In Table 3, the vertical text is hard to read. You could consider putting some numbers or abbreviations and reporting the text (horizontally) in the caption of the table

- The dependent variable of the regression analysis is the existence of suicidal ideation, but it is unclear how many participants answered positively to the question. This information is relevant. Please add.

Discussion

- As reported above, the contents are valid, but the overall readability is poor. Please, improve.

- Some sentences, such as the one in lines 280-281, are hard to follow. Please, revise.

- It is inappropriate to discuss results you do not present because that could induce the reader to think you did selective reporting. So then, the sentence between lines 289 and 292 should be removed, or their respective result should be added to the main text or in the appendix.

- Your discussion lacks limitations. Some are mentioned above. Please, revise accordingly.

Author Response

Dear reviewer,

Thank you very much for all your comments, questions, and suggestions! In the following pages you can find our point-by-point responses to your questions and comments. We hope that the revisions to the manuscript and our accompanying responses will be sufficient to make our manuscript suitable for publication!

Comments and Suggestions for Authors

The present manuscript is a national survey conducted in Romania, in which the authors attempt to identify variables predicting which patients are at high risk for suicidal ideation. The manuscript does not present severe English language mistakes, but the overall readability and style must be improved. The sentences in the introduction and discussion are poorly connected. Connecting conjunctions between sentences are missing, making the text lack smoothness. I suggest the authors ask for support from an English native speaker. Moreover, the introduction and the discussion might be shortened. Here are my requests:

We have improved the ‘overall readability and style’ of English language.

The sentences in the introduction and discussion are now better connected.

The introduction and the discussion were shortened.

Methods

- Please provide a reference to the CATI method

The following reference documents the CATI method. We have added it in the manuscript and in the list of references:

Kelly, J. (2008). Computer-assisted telephone interviewing (CATI). In P. J. Lavrakas (Ed.), Encyclopedia of survey research methods (pp. 123-125). Sage Publications, Inc., https://dx.doi.org/10.4135/9781412963947.n83.

- Please better describe the procedure for weighting, in statistical analysis:

Below we have provided a more detailed explanation of the weighting procedure. We have also inserted in the new version of the manuscript some more explanations about the sampling:

‘This study had a cross-sectional quantitative research design. Data were collected between June 28 and 30, 2021, using the Computer Assisted Telephone Interviewing [CATI] method (Kelly, 2008) (ZZ) Data gathering was conducted by the Romanian Institute for Evaluation and Strategy. A sample of 1102 individuals aged 18 and over, representative for the non-institutionalized adult population residing in Romania, were contacted using the random digit dialing (RDD) method. Data were weighted by gender, age, region, area of residence and level of education against the most recent official data available (the 2019 population estimates provided by the National Institute of Statistics in Romania, plus level of education data from the 2011 census), using the rake weighting procedure with 20 iterations and a delta of 0.5 in SPSS 22.’

- Please better describe if you applied or not multiple comparison corrections, and if not, report it as a limit and try to justify this choice

If the question relates to the results of correlations, and more precisely at the significance levels, we did not set a particular significance value of p (such as 0.01 or 0.05 for one-tailed or two-tailed hypotheses). Instead, in the Tables 1, 2 and 3, we have reported the level of significance calculated in the statistical program following the application of the Pearson correlation test, which was 0.01. By mistake, in the legend of the Table 1, two values of p are indicated, but the second is placed there by mistake. Consequently, in the new version of the manuscript we have deleted the sentence: ‘Correlation is significant at the 0.05 level (2-tailed).’

- The reference you cite in line 178 (for The Short Scale of Socio-Emotional Loneliness) has been translated into Francoise, German, Dutch, Russian, Bulgarian, Georgian, and Japanese. Is there any non-reported validation in the Romanian language? Did you adopt the standard procedure for forward and backward translation if not? It should be clearly stated as a study limitation if you did not. Moreover, it would be precious if at least the Chronbach alpha of the scale in the present study is reported.

The Short Scale of Socio-emotional Loneliness (SSSEL) has been previously used in the ‘Generations and Gender Survey Wave 1’ (GGS) in 2005. The project was supported by United Nations Economic Commission for Europe (UNECE). You can find more details on the programme’s website: https://www.ggp-i.org/. The first wave of GGS questionnaire was applied in 26 European countries. In Romania, it has been applied by the National Institute of Statistics resulting a sample of 11986 cases. We have taken the SSSEL from this questionnaire. The items of SSSEL refer to the current experiences. The value of Cronbach's Alpha of the SSSEL in this national sample was 0.5.

When it comes to the reliability of the SSSEL in our study, we have calculated it. The result indicates a low value of Cronbach alpha (0.3), but this is in line with the specifications related to this coefficient mentioned on the website which present the methodological aspects of this instrument. We have selected two information which could clarify this issue, and more details can be found on the following website: https://home.fsw.vu.nl/tg.van.tilburg/manual_loneliness_scale_1999.html:

Psychometric properties. Typically, a scale reliability in the 0.80 to 0.90 range is observed (Cronbach's alpha, KR-20 or rho). The homogeneity of the scale varies across studies, with Loevingers' H typically in the 0.30 to 0.50 range (higher when mail questionnaires were applied than in face-to-face interviewing), which is sufficient, but not very strong. The scale is discussed in Shaver and Brennan (1991).’

‘Reviews

The Committee Test Affairs (Commissie Testaangelegendheden Nederland; COTAN) of the Netherlands Institute of Psychologists (Nederlands Instituut van Psychologen; NIP) assessed the quality of the Loneliness Scale on April 18, 2000 as follows (Documentatie van Tests en Testresearch in Nederland, 2000): I. Basic assumptions of test construction: good; IIa. Quality of the test material: sufficient; IIb. Quality of the manual: sufficient; III. Norms: sufficient; IV. Reliability: good; Va. Content validity: sufficient; Vb. Criterium validity: insufficient due to absence of research.’

Taking all these into consideration, we did not mention the reliability test’s result in the manuscript. However, if you still consider it necessary, we will specify it as suggested!

- Regarding the question of Suicidal Ideation and long-term sadness, I have two questions:

  1. What is the time frame you considered for the questions? Because from the text, those seem lifetime questions. If those are lifetime questions, how could you use the variables as the outcome variables of present suicidal ideation?

Thank you for your questions and comment! Your remark is very pertinent!

The questions about the psychosocial factors (health; nutrition; physical activities; recreational activities; wealth; affective life; sex life; family relationships; social relationships; and income) refered to the past 12 months. We added this specification in the new draft, as follows:

‘Considering the last 12 months, how do you rate different aspects of your life?’

Questions related to QOL (overal happiness and life satisfaction), suicidal thoughts, long-term sadness, and life antecedents (trauma; abuse; chronic illness; deception; depression; and self-harm) were lifetime questions.

The items of the Short Scale of Socio-emotional loneliness referred to the recent past, meaning the last two weeks.

Taking these timeframes into consideration, it is true that we cannot identify from the collected data the appearance order of the investigated aspects, namely if the evaluation of the psychosocial factors is prior to the answers related to suicide ideation (SI) and prolonged sadness (PS) or not. We did not intend to test any causality relationships, but to search if the investigated variables (SI and PL) were associated with certain determinants identified in literature (e.g., psychosocial factors, QOL indicators, adverse life events, and socio-emotional loneliness). 

Also, we reconsidered the covariates used in the regression model, excluding loneliness and the psychosocial factors. Table 4 in the updated manuscript contains the new regression model.

  1. On the other hand, if you defined a strict and recent timeframe, the question is linked to suicidal risk if the participant answered yes. So then, the second question is more ethical. How did you manage participants' suicidal ideation? Did you suggest participants contact a specific number (suicide prevention lifeline) or go to the hospital seeking help?

As already specified, suicidal thoughts and long-term sadness were lifetime questions, so it was not the case for emergency assistance.

Results

- The correlation in table 1 reports a 0.05 level of significance. The others 0.01. Is the difference based on something? Did you apply multiple comparison corrections for the following tables? In case, why did you not in the first one?

Please find above our answer to this question!

- There is some text in bold in the tables, but the reason is not clarified in the caption of the table. Please, clarify.

In the Tables 1, 2 and 3, the values in bold remained from a working version of the paper, so they remained underlined by error. In the new manuscript, all the correlation coefficients are edited in the regular style. In the Table 4, the numbers in bold reflect the regression coefficients which are statistically significant (a p level of significance lower than or equal with 0.05).

- In Table 3, the vertical text is hard to read. You could consider putting some numbers or abbreviations and reporting the text (horizontally) in the caption of the table

Thank you for your recommendations! Now Table 3 has a legend with the list of the socio-emotional scale items and the number of each item is shown in the table’s header.

- The dependent variable of the regression analysis is the existence of suicidal ideation, but it is unclear how many participants answered positively to the question. This information is relevant. Please add.

Following your request, we mentioned in the updated manuscript the weight of people who reported SI and PS (8% and 40%, respectively). You can find this information at the end of the sub-section ‘3.1. Descriptive statistics’ in the updated manuscript. It is a very important information indeed!

Discussion

- As reported above, the contents are valid, but the overall readability is poor. Please, improve.

We checked the English language, especially since the introduction and discussion sections have undergone substantial changes.

- Some sentences, such as the one in lines 280-281, are hard to follow. Please, revise.

We did it. For instance, in the ‘Discussion’ section, the indicated sentence (lines 280-281): ‘A higher impact of the socio-demographic risk factors for suicide was observed whenever they were less prevalent in the studied population [4]’, was revised as follows: ‘The impact of socio-demographic factors on developing suicidal behaviors was found to be negatively correlated with their prevalence in the studied population [4]’.

- It is inappropriate to discuss results you do not present because that could induce the reader to think you did selective reporting. So then, the sentence between lines 289 and 292 should be removed, or their respective result should be added to the main text or in the appendix.

Your comment is pertinent! Therefore, we have removed the text related to findings not mentioned in the ‘Results’ section (between lines 289 and 292 of the first submitted manuscript).

- Your discussion lacks limitations. Some are mentioned above. Please, revise accordingly.

This is a regrettable negligence! The new manuscript contains a paragraph related to the limits of the study, which includes a part of the sensitive issues you underlied in your review report!

We hope that our answers are satisfying! We stay at your disposal for any further clarifications related to the content of the new manuscript or to our answers at your questions and comments!

Submission Date

01 July 2022

Date of this review

11 Jul 2022 08:40:23

Sincerely yours,

Mihaela Laura Vica on behalf of the authors

Reviewer 2 Report

In my considered opinion the article "Determinants of suicide ideation in the Romanian population" examines a current issue, as it was revealed and argued (including with current statistics) by the authors in the introduction of the article. The article highlights through a quantitative approach the significance of the impact of psychosocial factors, quality of life and loneliness dimensions and adverse life antecedents on suicide ideation and prolonged sadness.

The article has a logical, coherent structure, which makes it easy for the reader to follow, with pertinent points of view selected from 39 bibliographic sources, relevant to the topic addressed, of which over 60% are from the last five years, without self-citation.

Each section is sufficiently clearly described and comprehensive. The methodology used is suitable for the purpose of the study, the research design being pertinent for testing the hypotheses. The sampling algorithm was sufficiently explained and appropriate.

In order to publish the article, we recommend the following:

- mention who are the authors of the survey questionnaire applied to the participants (was it the conception of the authors of this article or the survey questionnaire was developed by other authors, except "Short Scale of Socio-Emotional Loneliness"?);

- mention the source of the text written ad literam on lines 82-85; 91-96 (identified through the Ithenticate software as being similar to another paragraph published online) or revise the text;

- to develop the practical implications of the study results;

- to specify the limits of the research from the point of view of the methodology applied or other aspects if appropriate and specify possible directions of further research.

In conclusion, we recommend publishing the article with minor revisions because it has potential to be read not only by researchers, but also by the general public.

Author Response

Dear reviewer,

Thank you very much for all your comments, questions, and suggestions! Below you can find our point-by-point responses to your questions and comments. We hope that the revisions to the manuscript and our accompanying responses will improve the quality of the paper and will make it more suitable for publication!

Comments and Suggestions for Authors

In my considered opinion the article "Determinants of suicide ideation in the Romanian population" examines a current issue, as it was revealed and argued (including with current statistics) by the authors in the introduction of the article. The article highlights through a quantitative approach the significance of the impact of psychosocial factors, quality of life and loneliness dimensions and adverse life antecedents on suicide ideation and prolonged sadness.

The article has a logical, coherent structure, which makes it easy for the reader to follow, with pertinent points of view selected from 39 bibliographic sources, relevant to the topic addressed, of which over 60% are from the last five years, without self-citation.

Each section is sufficiently clearly described and comprehensive. The methodology used is suitable for the purpose of the study, the research design being pertinent for testing the hypotheses. The sampling algorithm was sufficiently explained and appropriate.

In order to publish the article, we recommend the following:

- mention who are the authors of the survey questionnaire applied to the participants (was it the conception of the authors of this article or the survey questionnaire was developed by other authors, except "Short Scale of Socio-Emotional Loneliness"?);

Thank you very much for your comment!

The set of 10 aspects of life measuring different psychosocial determinants (health; nutrition; physical activities; recreational activities; wealth; affective life; sex life; family relationships; social relationships; and income) was adapted after the Devins Illness Intrusiveness Rating Scale (see the reference below). It is mentioned at the ‘Variables and measurement’ section in the updated version of the manuscript.

Devins G (1994b). Illness intrusiveness and the psychosocial impact of lifestyle disruptions in chronic life – threatening disease. Adv Ren Repl Ther, 1: 251–263.

We have also added a relevant reference for the two quality of life indicators in the same section.

Regarding the set of life antecedents (trauma; abuse; chronic illness; deception; depression; and self-harm), they were introduced in the questionnaire by the principal investigator after carrying out a small scale qualitative research, consisting of few narrative interviews with psychiatric inpatients who experienced suicide attempts. The unpublished findings of those interviews, together with the study of the literature helped the principal investigator to distinguish a set of adverse life events relevant to the topic of the study.

- mention the source of the text written ad literam on lines 82-85; 91-96 (identified through the Ithenticate software as being similar to another paragraph published online) or revise the text;

Thank you for your observation! The fragments in red were all depicted from the WHO document. By mistake we omitted to mark the final paragraph as a quote from reference:

https://apps.who.int/iris/bitstream/handle/10665/67215/%20WHO_NMH_NPH_02.8.pdf?sequence=1&isAllowed=y.

As a result of the massive reorganization of the ‘Introduction’ and ‘Discussion’ sections, we have decided to remove these ideas, as they were not sufficiently connected to the investigated topic of our study.

- to develop the practical implications of the study results;

Thank you for your suggestion! We proposed a set of recommendation for intervention and prevention in the concluding section.

- to specify the limits of the research from the point of view of the methodology applied or other aspects if appropriate and specify possible directions of further research.

Thank you for your comment! The lack of limits was a regrettable negligence! We have added a paragraph related to methodological issues! Also, possible further directions of research about suicide ideation and behaviours can be imagined.

In conclusion, we recommend publishing the article with minor revisions because it has potential to be read not only by researchers, but also by the general public.

Submission Date

01 July 2022

Date of this review

26 Jul 2022 21:54:32

Sincerely yours,

Mihaela Laura Vica on behalf of the authors

Reviewer 3 Report

This is an important and interesting study because we lack data on the prevalence of suicidal thoughts and intentions in the general population. I am glad that the authors took up this task. They did it rather well, but there are some issues that need to be corrected.

  1. „Increases in depression symptoms and suicide-related outcomes among US adolescents over thelast decade were linked to social media screen-time” I believe that this sentence is an over-interpretation of the quoted publications in which there is no direct relationship between suicidal behavior and the social media. This should be corrected.

  2. I think the Introduction is too extensive, describing important facts, but it goes beyond the aims of the publication. This section should be shortened.

  3. Unfortunately, the tables are not eligible and difficult to read due to the selection of fonts and their size. Please kindly correct it.

  4. Could you also give confidence intervals for logistic regression results (Table 4)

  5. There are too many side threads in the discussion, including clinical ones, unnecessarily extending the topic. For example, the study does not investigate the clinical diagnosis of depression or current suicidal intentions. Please concentrate on discussing main findings

  6. Please discuss limitations of the study

  7. Most of the correlation analysis results do not exceed a score of 0.3, which means weak correlation, not strong. Please discuss this fact.

  8. Did the telephone interviewers have any psychological training? It is not easy to assess suicidal intentions and prolonged sadness using only simple questions posed in the methodology section. This is a potential bias source.

  9. Data analysis - what statistical tests were used?

  10. Since the authors state that this is a representative sample of the population, can you please assess the epidemiological prevalence of suicide intentions and prolonged sadness in various socio-demographic groups? This would be a very important knowledge for public health.

Author Response

Dear reviewer,

Thank you very much for all your comments, questions, and suggestions! Please find below our point-by-point responses to your questions and comments. We hope that the revisions to the manuscript and our accompanying responses will be sufficient to make our manuscript suitable for publication!

Comments and Suggestions for Authors

This is an important and interesting study because we lack data on the prevalence of suicidal thoughts and intentions in the general population. I am glad that the authors took up this task. They did it rather well, but there are some issues that need to be corrected.

„Increases in depression symptoms and suicide-related outcomes among US adolescents over the last decade were linked to social media screen-time” I believe that this sentence is an over-interpretation of the quoted publications in which there is no direct relationship between suicidal behavior and the social media. This should be corrected.

Thank you for your comment! We have removed the indicated sentence in the updated manuscript. Instead, we have extracted from the same source a finding which is more connected with the results of our study:

‘As for the youngsters, extensive social media exposure in the internet era is held accountable for the growing social isolation and loneliness, for youngsters in particular [24].’

I think the Introduction is too extensive, describing important facts, but it goes beyond the aims of the publication. This section should be shortened.

Thank you for your pertinent comment! In the updated manuscript, we have modified the ‘Introduction’ and ‘Discussion’ sections in order to be more connected to each other and with the findings of our study. The introduction has been shortened, as suggested.

Unfortunately, the tables are not eligible and difficult to read due to the selection of fonts and their size. Please kindly correct it.

Thank you for your observation! We have tried to improve the aspect of the tables and to homogenize the texts in the Tables 1, 2 and 3. We made Table 3 easier to read. Thus, we put numbers and abbreviations for the items of the Socio-emotional loneliness scale and reported the text (horizontally) in the caption of the table.

Could you also give confidence intervals for logistic regression results (Table 4)

Yes. We have added two columns in the Table 4 with the requested data.

There are too many side threads in the discussion, including clinical ones, unnecessarily extending the topic. For example, the study does not investigate the clinical diagnosis of depression or current suicidal intentions. Please concentrate on discussing main findings

Thank you for your comment and request! We modified the discussions section in order to be consistent with the findings of our study.

Please discuss limitations of the study

We have introduced a paragraph relating to the limits of our study at the end of the updated version of the manuscript.

Most of the correlation analysis results do not exceed a score of 0.3, which means weak correlation, not strong. Please discuss this fact.

It is correct. However, taking into account the size of the sample, even a lower value of the Pearson correlation coefficient can be considered substantial and strongly significant. We found this explanation in the following source:

Weinbach R.W., Grinnell R.M. Jr., 1987 - Statistics for Social Workers, Longman Inc. New York & London.

Did the telephone interviewers have any psychological training? It is not easy to assess suicidal intentions and prolonged sadness using only simple questions posed in the methodology section. This is a potential bias source.

Telephone interviewers had no psychological training. Also, no professional assessments were conducted in regard to the suicidal intentions or prolonged sadness. Statistical analyses were based on the respondents’ self-assessment alone.

Data analysis - what statistical tests were used?

We have applied the Pearson correlation test and the binary logistic regression.

Since the authors state that this is a representative sample of the population, can you please assess the epidemiological prevalence of suicide intentions and prolonged sadness in various socio-demographic groups? This would be a very important knowledge for public health.

Thank you for your comment and suggestion! Thank you for your recommendation! We appreciate it very much! In the updated version of the manuscript, we mentioned the weight of people who reported suicide ideation (SI) and prolonged sadness (PS): 8% and 40%, respectively. You can find this information at the end of the sub-section ‘3.1. Descriptive statistics’ in the updated manuscript. It is a very important information indeed! You can find the requested data in the table below:

Socio-demographic variables

SI (%)

PS (%)

Sex

Male

33

36

Female

67

64

Area of residence

Urban

52

54

Rural

48

46

Education

Primary or lower education

62

46

Secondary education

29

44

Higher education

9

10

Age group

18-35

30

26

36-50

31

25

51-65

28

20

Over 65

11

29

Submission Date

01 July 2022

Date of this review

26 Jul 2022 13:16:45

Sincerely yours,

Mihaela Laura Vica on behalf of the authors

Round 2

Reviewer 3 Report

I am pleased with the corrections included in revised version of the manuscript. I would like to thank the authors for this interesting research.